# Effect of Food Waste Condensate Concentration on the Performance of Microbial Fuel Cells with Different Cathode Assemblies

Theofilos Kamperidis [1], Pavlos K. Pandis [1], Christos Argirusis [1], Gerasimos Lyberatos [1,2] and Asimina Tremouli [1,*]

[1] School of Chemical Engineering, National Technical University of Athens, 157 72 Athens, Greece; kamp.theo@gmail.com (T.K.); ppandis@chemeng.ntua.gr (P.K.P.); amca@chemeng.ntua.gr (C.A.); lyberatos@chemeng.ntua.gr (G.L.)

[2] Institute of Chemical Engineering Sciences (ICE-HT), 265 04 Patras, Greece

* Correspondence: atremouli@chemeng.ntua.gr

**Abstract:** The aim of this study is to examine the effect of food waste condensate concentration (400–4000 mg COD/L) on the performance of two microbial fuel cells (MFCs). Food waste condensate is produced after condensing the vapors that result from drying and shredding of household food waste (HFW). Two identical single-chamber MFCs were constructed with different cathodic assemblies based on GoreTex cloth (Cell 1) and mullite (Cell 2) materials. Linear sweep voltammetry (LSV) and electrochemical impedance spectroscopy (EIS) measurements were carried out to measure the maximum power output and the internal resistances of the cells. High COD removal efficiencies (>86%) were observed in all cases. Both cells performed better at low initial condensate concentrations (400–600 mg COD/L). Cell 1 achieved maximum electricity yield (1.51 mJ/g COD/L) at 500 mg COD/L and maximum coulombic efficiency (6.9%) at 400 mg COD/L. Cell 2 achieved maximum coulombic efficiency (51%) as well as maximum electricity yield (25.9 mJ/g COD/L) at 400 mg COD/L. Maximum power was observed at 600 mg COD/L for Cell 1 (14.2 mW/m$^2$) and Cell 2 (14.4 mW/m$^2$). Impedance measurements revealed that the charge transfer resistance and the solution resistance increased significantly with increasing condensate concentration in both cells.

**Keywords:** MFC; food waste condensate; mullite; GoreTex; cathode assembly; electrochemical impedance spectroscopy

## 1. Introduction

Biowaste mainly consists of food waste (60%) and garden waste and corresponds to a share of 34% of the total municipal waste in the EU. Only in the EU-28, 88 million tons of food are wasted every year [1]. Given the consequences of biowaste disposal as well as its potential for valorization, biowaste has to be considered more as a valuable resource stream than as a waste to dispose of. Usually, disposal of biowaste, such as food waste, includes landfilling or incineration. However, these approaches have a severe impact on the environment. Landfills cause leachate formation and subsequent groundwater pollution, as well as greenhouse gas emissions [2]. On the other hand, incineration consumes high amounts of energy and produces high amounts of carbon dioxide emissions [3]. Given the fact that almost 53% of the total food waste produced in the European Union corresponds to household food waste (HFW), along with the fact that HFW is a material rich in carbon and nitrogen, different approaches have emerged using HFW as a feedstock [4,5], for the production of energy and value-added products such as biogas, biosurfactants, bioplastics and organic fertilizers [4,6], as adsorbents for the removal of dye effluents from water streams [2] and as platform chemicals such as lactic acid [7,8].

In order to fully exploit HFW, sorting at the source needs to be carried out before its use as a feedstock. Separate collection from households ensures that plastic, metal, glass

and other inorganic materials are not mixed with HFW [9]. Moreover, HFW, due to its high-water content and its complex organic matter, has the tendency to be spontaneously biodegraded by aerobic and/or anaerobic microorganisms emitting odors and raising health and environmental concerns [10]. Under the view of confronting these drawbacks and valorizing the valuable material of HFW, an alternative approach has been recently developed within the framework of the Horizon 2020 Waste4Think Project [11]. During this Project, in the Municipality of Chalandri, Greece, volunteering households along with local food markets sorted their food waste from the rest of their waste streams. Moreover, in order to overcome the major issue of HFW spontaneous biodegradation, the collected HFW was dried and shredded, thus producing a homogenized solid biomass with low moisture content so that it might be stored for prolonged periods of time. The main product from the drying process, named food residue biomass (FORBI), is rich in carbon and nitrogen, making it an ideal substrate for anaerobic processes [12]. Consequently, FORBI has been examined in different bioprocesses. Specifically, it was used as a substrate for dark fermentation [13], as a feedstock in a periodic anaerobic baffled reactor (PABR) [14] and as a substrate for electricity production in MFCs [15]. FORBI was also examined for biofuel production (biohydrogen, bioethanol and methane) [10].

However, along with FORBI production, during the drying process approximately 75–80% of the moisture contained in HFW is removed and the vapors are collected in the condenser [12]. The produced liquid, named condensate, is a material with high organic and low nitrogen contents, making it a potential feedstock for different processes [12]. Under the scope of fully exploiting all the by-products from the drying process, Lytras et al. (2020) examined the approach of co-digesting the condensate with waste-activated sludge (WAS). In particular, the condensate, due to its low nitrogen content, was co-digested with WAS in a CSTR with 100 L of working volume [12]. The condensate to WAS ratio was 1:5, resulting in an average methane percentage of 74.3% in the produced biogas while the overall methane production was 343 mL methane/g COD [12].

The microbial fuel cell (MFC) is a technology that has exhibited the ability to process a wide spectrum of substrates [16]. MFCs utilize electrochemically active bacteria that oxidize an organic substrate and reduce an electron acceptor, resulting in simultaneous wastewater treatment and current production [17]. In order to maximize MFC performance, different anode and cathode assemblies have been tested. Ideally, an electrode should offer high conductivity, electrochemical stability and biocompatibility with high specific surface area if it comes in contact with microorganisms [18]. In this direction, several carbon-based materials have been examined as electrodes (such as carbon paper, carbon felt, graphite granules, graphite rod, graphite sheet and activated carbon) [19]. In order to avoid short circuits, a separator is required between the anode and the cathode. Conventional membrane separators (e.g., proton exchange membranes), however, have a high cost [20] and increase ohmic losses [17], limiting MFC performance efficiency [21]. Ceramic materials can be used both as separators and as structural material in MFC systems. The use of ceramic materials in MFCs is an option that offers advantages, such as resistance to fouling [22], lower cost when compared to other materials used (e.g., platinum-coated electrodes) [23] and thermal, chemical and mechanical stability [24]. Ceramic separators coated with oxygen reduction catalyst have been successfully used in single-chamber MFCs [25,26].

Under the concept of the circular economy of HFW exploitation, this paper's novelty lies in utilizing the food waste condensate for electricity production using MFC technology while it is simultaneously treated. For this goal two single-chamber air cathode MFCs with different cathode assemblies were tested. In particular, GoreTex cloth and ceramic material (mullite) were assessed using $MnO_2$ as the catalyst in both systems. In addition, the effect of different initial condensate concentrations on MFC performance was examined. In order to assess the effect of the initial condensate concentration as well as the effect of the different cathode assemblies on the internal resistance of the units, a detailed electrochemical characterization was performed.

## 2. Materials and Methods

### 2.1. MFCs Set Up and Operation

Two single-chamber membraneless MFCs similar to [14] and [27], respectively, were constructed for this work. In the case of Cell 1, four perforated Plexiglas tubes were wrapped with GoreTex cloth coated with electrocatalytic paste. The paste was made by mixing 12 g graphite paint (YSHIELD HSF54), 3 g $MnO_2$ (EMD, TOSOH HELLAS), 3 mL ethanol and 3 mL xylene [28]. Cell 2 was constructed with the same paste, which was applied with a brush coat technique on the inner surface of four mullite tubes with 20% open porosity. In Figure 1, one GoreTex electrode and one mullite electrode are presented. The cathodic materials were selected by choosing low-cost materials that require little to no processing and whose performance has already been tested in the MFC technology.

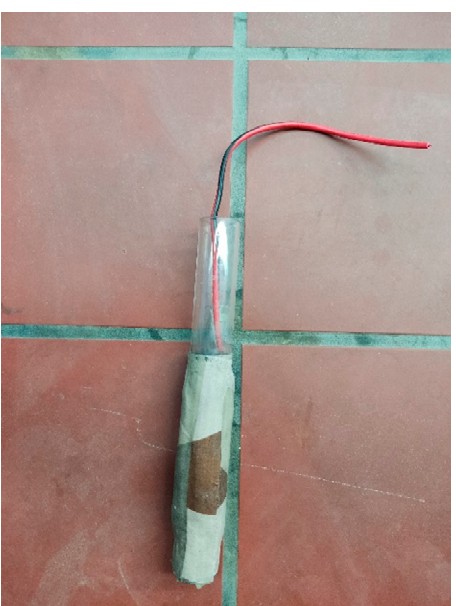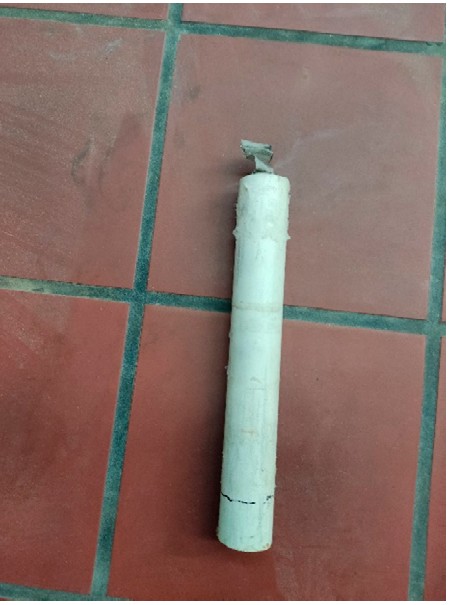

**Figure 1.** Images of cathodic electrodes, GoreTex electrode (**left**) with $MnO_2$ paste and mullite electrode (**right**) with the same $MnO_2$ paste.

The anodic set-up was identical for both MFCs. In particular, graphite granules (250 gr) were used as the anode electrode and a graphite rod was inserted in the anode chamber for the electron collection. Copper wire was used for electrode connection. The external resistance was set at 100 Ω (except when stated otherwise) in both units. The two cells were placed in a temperature-controlled room at 27 °C.

The anodic liquid volume was 150 mL. The units were operated in batch mode and were acclimated using glucose as substrate (1000 mg COD/L), similar to [15]. Specifically, during the first three batch cycles, anaerobic sludge (10% *v/v*) was inserted in the anolyte. Anaerobic sludge was obtained from the Likovrisi (Athens, Greece) sewage treatment plant. The characteristics of the anaerobic sludge obtained are presented in Table 1:

**Table 1.** Characteristics of anaerobic sludge used for the acclimation of the MFC units.

| Anaerobic Sludge | pH | Conductivity (mS/cm) | Soluble COD (mg/L) | Total COD (mg/L) | TSS (g/L) | VSS (g/L) |
|---|---|---|---|---|---|---|
| | 7.3 | 4.4 | 470 | 21,900 | 49 | 21 |

Following the acclimation period, the synthetic wastewater was switched with condensate originating during the drying process of HFW. HFW was collected door-to-door from households in the Municipality of Chalandri, Athens. Condensate was produced by a

dryer/shredder GAIA GC-300. The dryer/shredder was loaded with 120 kg of pre-sorted HFW. The drying took place at 172 °C for 9 h, with simultaneous shredding of the HFW. Two products were generated from the drying process: the solid homogenous biomass product called FORBI (Food Residue Biomass) [29] and vapors. By condensing the vapors, liquid condensate was produced [12]. The main characteristics of the condensate were 13 g COD/L, pH 3.5 ± 0.4, while the conductivity was 262 ± 100 μS/cm. Before feeding the condensate to the anode chamber, it was mixed with phosphate buffer (5.29 g/L $NaH_2PO_4 \cdot 2H_2O$, 3.45 g/L $Na_2HPO_4 \cdot 2H_2O$) and potassium chloride (0.16 g/L) (similar to [15]). Following the addition of the phosphate buffer and potassium chloride, the average pH of the anolyte was 6.2 ± 0.7 and the average conductivity was 5 ± 0.6 mS/cm (Table 2). The same initial condensate concentrations were examined in both cells, in order to compare each cell's operation performance with different separators. The condensate initial concentration (400–4000 mg COD/L) was increased in consecutive operation cycles. For each initial condensate concentration, two operation cycles were carried out. The characteristics of the condensate feed are presented in detail in Table 2:

**Table 2.** Characteristics of the condensate feed after it was mixed with the buffer solution.

| Cycle Number | mg COD/L | pH | Conductivity (mS/cm) |
| --- | --- | --- | --- |
| 1st | 400 | 6.7 | 5.9 |
| 2nd | 500 | 6.9 | 5.0 |
| 3rd | 600 | 6.6 | 5.0 |
| 4th | 800 | 6.5 | 4.9 |
| 5th | 1200 | 6.6 | 4.9 |
| 6th | 1400 | 6.6 | 5.2 |
| 7th | 3000 | 5.1 | 5.4 |
| 8th | 4000 | 4.8 | 3.8 |

### 2.2. Analytical Methods, Calculations and Electrochemical Characterization

The voltage of the two cells was recorded at 2 min intervals by a Keysight LXI Data Acquisition system. The measurements of pH and conductivity were conducted using digital instruments WTW INOLAB PH720 and WTW INOLAB, respectively. The soluble COD was measured according to [30].

In order to further assess the performance of the two MFCs and the condensate as an electron donor, the coulombic efficiency (*CE*) of the cells and the electricity yield (*El*$_{yield}$) were calculated. *CE* is defined as the fraction of the charge produced to the total charge contained in the substrate, and is calculated by the equation:

$$CE = \frac{MO_2 \int_0^t I dt}{F \cdot b \cdot V \cdot \Delta COD} \tag{1}$$

where $MO_2$ is the molecular weight of oxygen (32 g/mol), I is the current generated during the operation cycle (A), *F* is the Faraday constant (46,985 C/mol), *b* is the number of electrons participating in the reaction (4), *V* is the working volume of the cells (150 mL) and $\Delta COD$ is the consumed *COD* ($C_{Initial}$–$C_{Final}$, g COD/L).

The electricity yield (mJ/g COD/L) per g/L of initial *COD* concentration was calculated by the equation:

$$El_{yield} = \frac{\int_0^t P dt}{COD_{condensate}} \tag{2}$$

Here *P* is the power generated (in W) during each cycle and $COD_{Condensate}$ is the initial *COD* concentration of each cycle (in mg COD/L).

Linear sweep voltammetry (LSV) and electrochemical impedance spectroscopy (EIS) experiments were carried out using a Potentiostat–Galvanostat (PGSTAT128N—AUTOLAB) with an Ag/AgCl reference electrode. The electrochemical experiments were conducted at the beginning of each operation cycle, after the feeding of the cells. Before the electrochemi-

cal experiments, each cell achieved open-circuit voltage (OCV) by removing the external resistance. LSV was conducted from OCV to short circuit with a negative step (0.005 mV/s) in order to estimate the maximum power output of each cell. EIS measurements estimated the internal resistance of each cell with a frequency range of 1 MHz–1 mHz using a stimulus of 10 mV amplitude. The overall performance of the cell was estimated using the anode as the working electrode and the cathode as the counter electrode, and the three-electrode set-up was completed with a reference (Ag/AgCl) electrode in connection with the counter. The electrical equivalent circuit for the EIS fit was similar to [31,32], calculating the internal resistances of the cell RS, RBF and RCT (S: solution; BF: biofilm; and CT: charge transfer resistances).

### 3. Results and Discussion

*3.1. MFC Operation at Different Initial Condensate Concentrations*

Following the inoculation period using synthetic wastewater with glucose, condensate was used as substrate. Different condensate concentrations were examined in the range of 400 mg COD/L to 4000 mg COD/L. The current output and the COD concentration versus time for the different initial condensate concentrations for Cells 1 and 2 are shown in Figures 1 and 2, respectively.

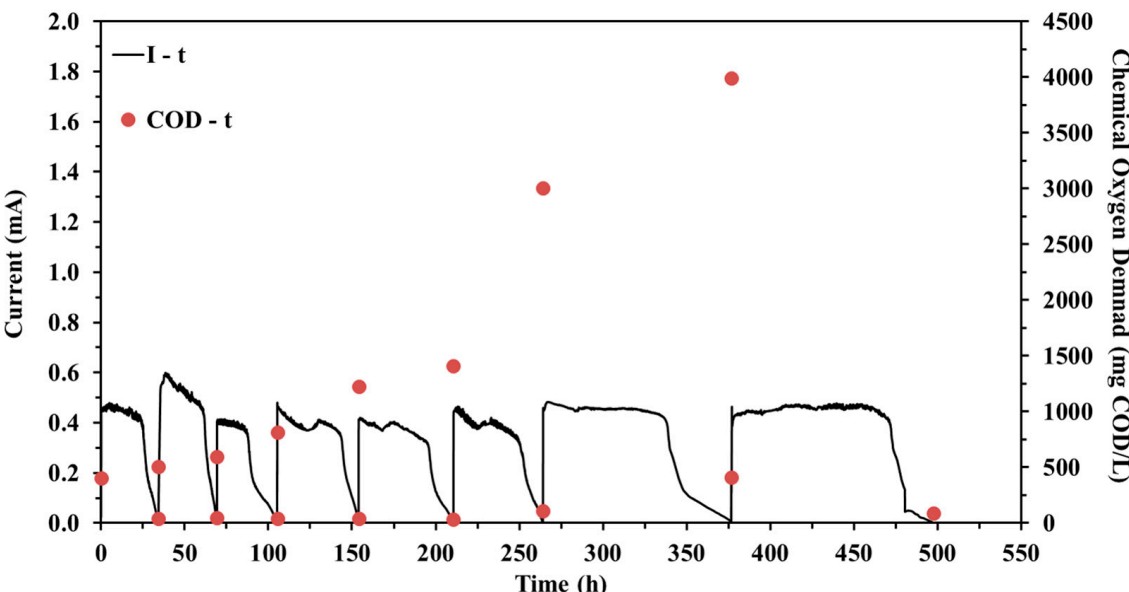

**Figure 2.** Voltage output (**left**) and COD concentration (**right**) versus time of Cell 1 (GoreTex with MnO$_2$ as cathodic electrodes).

As can be seen from Figures 2 and 3, the COD removal efficiency was high for all initial concentrations (>86%). In the case of Cell 1 (GoreTex cloth separator), the maximum current output (Imax) for all cycles was approximately 0.5 mA except in the case of 500 mg COD/L, where the maximum current output was somewhat higher (0.6 mA) (Figure 4). Although the COD removal efficiency and the current output remained relatively stable with increasing initial condensate concentration, the duration of cycles, the CE and the $El_{yield}$ were affected by the initial organic load. The duration of the cycles was increased as the initial condensate concentration increased (Figures 2 and 3). A similar observation was made by [32], increasing the initial COD in a two-chamber MFC. The electricity yields for all cycles, as calculated by Equation (2) are presented in detail in Table 3 and the coulombic efficiencies (CE) as calculated by Equation (1) are presented in Figure 4. In particular, the duration of each cycle increased from 34 h to 121 h, as the initial condensate concentration increased from 400 mg COD/L to 4000 mg COD/L. On the contrary, *CE* and $El_{yield}$ were decreased by 67% and 60%, respectively, when the initial condensate

concentration gradually increased from 400 mg COD/L to 4000 mg COD/L, (Figure 4). Specifically, the maximum CE (4.3%) was obtained at 400 mg COD/L, indicating that Cell 1 operated better at lower initial condensate concentrations. Similarly, the maximum $El_{yield}$ was achieved at 400 and 500 mg COD/L and was equal to 1.25 and 1.54 mJ/g COD/L, respectively (Table 3 and Figure 4). This behavior may be attributed to the decrease in the anolyte's pH and conductivity as the initial condensate concentrations increased (400 mg COD/L: pH 6.7, 5.9 mS/cm to 4000 mg COD/L: pH 4.8, 3.8 mS/cm) (Table 2). It is known that electrogenic bacteria perform better at an environment close to neutral pH [33], whereas low conductivity values also limit the performance of the cells [27]. Additionally, although relatively low CE values were achieved, high COD removal efficiency was obtained, indicating that antagonistic microorganisms consumed a high portion of the organic material [34].

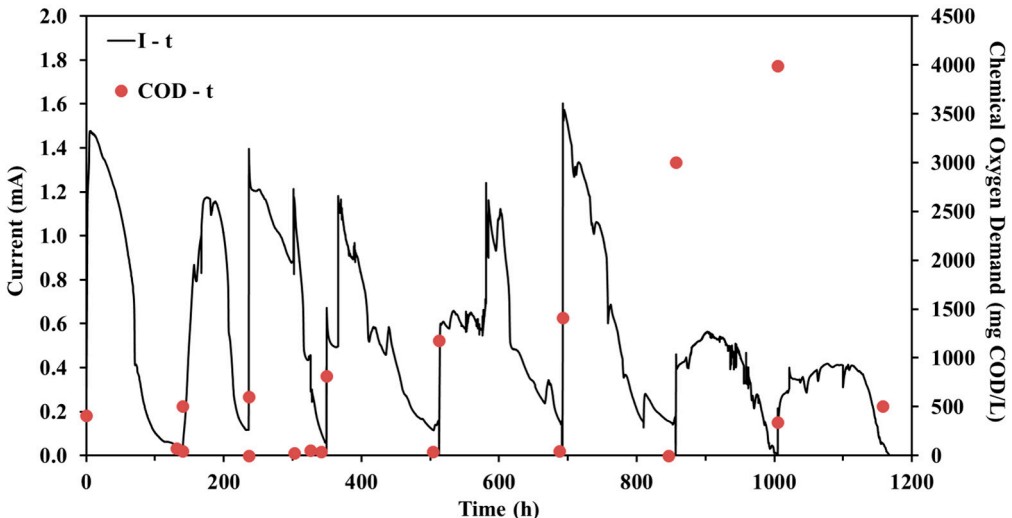

**Figure 3.** Voltage output (**left**) and COD concentration (**right**) versus time of Cell 2 (mullite with MnO$_2$ as cathodic electrodes).

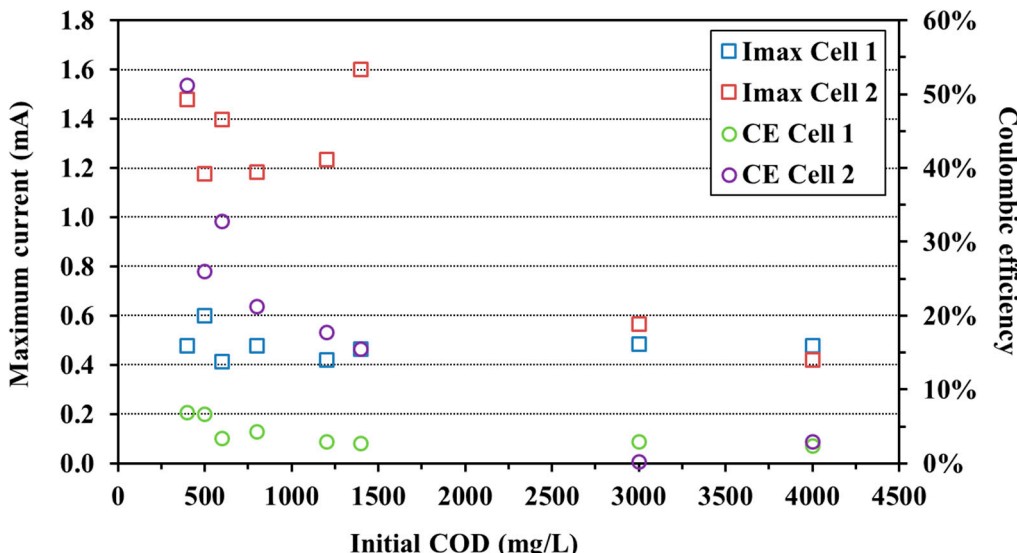

**Figure 4.** Maximum current (**left**) and Coulombic efficiency (**right**) versus the initial COD concentrations of condensate fed to the two cells.

**Table 3.** Operational characteristics (maximum current output, cycle duration, COD removal efficiency, coulombic efficiency and electricity yield) were achieved for Cell 1 and Cell 2 at different initial condensate concentrations.

| Initial Condensate Concentration (mg COD/L) | Δt Cycle (h) | | COD Removal | | Elyield (mJ/g COD/L) | |
|---|---|---|---|---|---|---|
| | Cell 1 | Cell 2 | Cell 1 | Cell 2 | Cell 1 | Cell 2 |
| 400 | 34 | 140 | 90% | 89% | 1.31 | 25.9 |
| 500 | 35 | 96 | 91% | 99% | 1.51 | 12.2 |
| 600 | 36 | 113 | 93% | 93% | 0.53 | 15.3 |
| 800 | 49 | 164 | 95% | 95% | 0.77 | 6.8 |
| 1200 | 57 | 179 | 97% | 96% | 0.51 | 5.8 |
| 1400 | 54 | 164 | 92% | 99% | 0.46 | 7.9 |
| 3000 | 113 | 148 | 86% | 89% | 0.53 | 0.9 |
| 4000 | 121 | 152 | 98% | 87% | 0.50 | 0.5 |

In the case of Cell 2 (mullite separator), the maximum current output values that were achieved were two-to-three-fold higher when compared with Cell 1. In particular, Imax ranged between 1.2 and 1.6 mA for the initial condensate concentrations of 400 to 1400 mg COD/L, whereas Imax was 0.6 mA and 0.4 mA for the 3000 and 4000 mg COD/L initial concentrations, respectively (Figure 4). Additionally, the duration of each cycle was approximately three-to-four-fold longer than the corresponding Cell 1 cycles, except in the case of the higher initial condensate concentrations (3000 and 4000 mg COD/L), where similar cycle durations were observed for both cells (Table 3). The duration of the cycles ranged between 96 h and 179 h, and on the contrary, with Cell 1 the duration of each cycle did not increase with the initial COD concentration increase. Similarly to Cell 1, CE and El$_{yield}$ (calculated by Equation (1) and Equation (2), respectively) were decreased by 96% and 98% when the initial condensate concentration gradually increased from 400 mg COD/L to 4000 mg COD/L, respectively (Table 3 and Figure 4). In particular, the maximum CE (51%) was obtained at 400 mg COD/L. Similarly, the maximum Elyield was achieved at 400 mg COD/L and was 26.5 mJ/g COD/L. Although both cells achieved high COD removal efficiency values, it is clear in both cases that the decrease in the anolyte's pH and conductivity with the initial condensate concentration increase deteriorates the cell's performance. Moreover, the performance of the cells in terms of electricity production is more efficient using the mullite separator instead of the GoreTex cloth.

*3.2. Linear Sweep Voltammetry Experiments at Different Initial Condensate Concentrations*

Linear sweep voltammetry (LSV) experiments were conducted in order to examine the effect of condensate concentration on the polarization performance of the cells. Polarization curves were obtained for initial concentrations of 600, 800, 1400 and 3000 mg COD/L. Figures 5 and 6 present the results of the LSV experiments, voltage and power density versus current density for Cell 1 and 2, respectively. Power density is calculated by normalizing the measured power to the surface of the cathodic electrodes (Cell 1: 58 cm$^2$ and Cell 2: 250 cm$^2$).

As can be seen from Figure 5, the maximum power density (P$_{max}$ 14.2 mW/m$^2$) was achieved at 600 mg COD/L. Additionally, the increase in the initial condensate concentration led to a gradual decrease in P$_{max}$ values (600 mg COD/L: 10.9 mW/m$^2$, 800 mg COD/L: 12.4 mW/m$^2$, 1400 mg COD/L: 10.1 mW/m$^2$, 3000 mg COD/L: 6 mW/m$^2$). The lowest maximum power output (6 mW/m$^2$) was obtained at 3000 mg COD/L. Based on the voltage versus current density lines (Figure 5), in all cases, ohmic losses dominated in Cell 1. The slope of the voltage versus current density lines indicates the internal resistances, which are greater the steeper the slope. The internal resistances of Cell 1 which are calculated using the power density peak method also increased with increasing initial condensate concentration (600 mg COD/L: 848 Ω, 800 mg COD/L: 956, 1400 mg COD/L: 1016 Ω, 3000 mg COD/L: 1365 Ω).

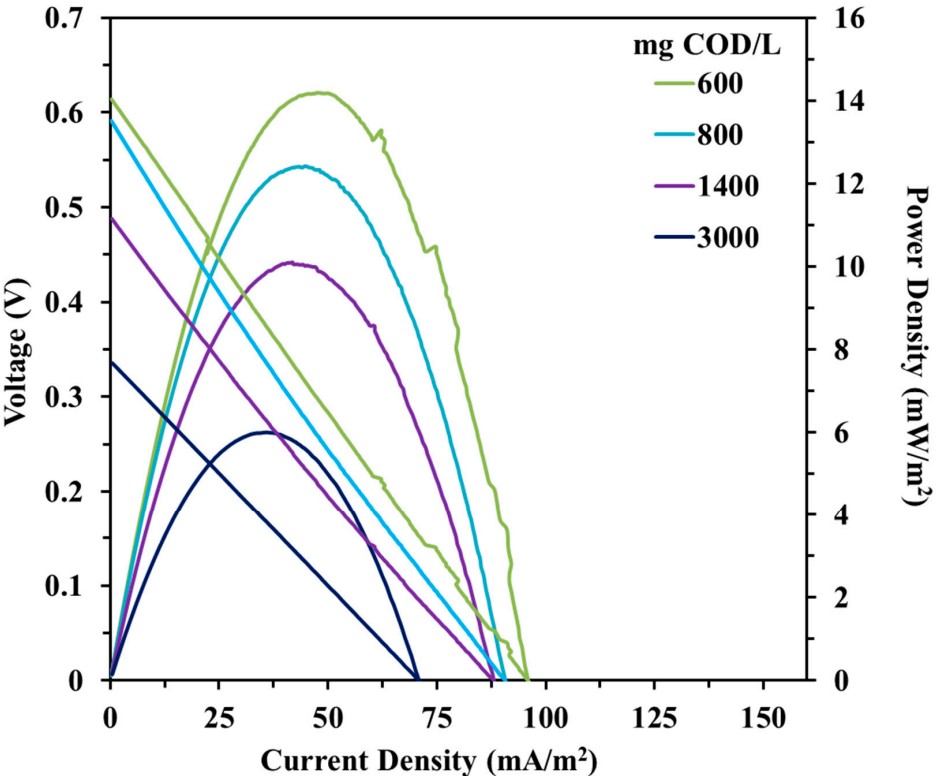

**Figure 5.** Power density versus current density as extracted by LSV experiments on Cell 1 (GoreTex assembly) for the following initial condensate concentrations (mg COD/L): 600 ▬, 800 ▬, 1400 ▬ and 3000 ▬.

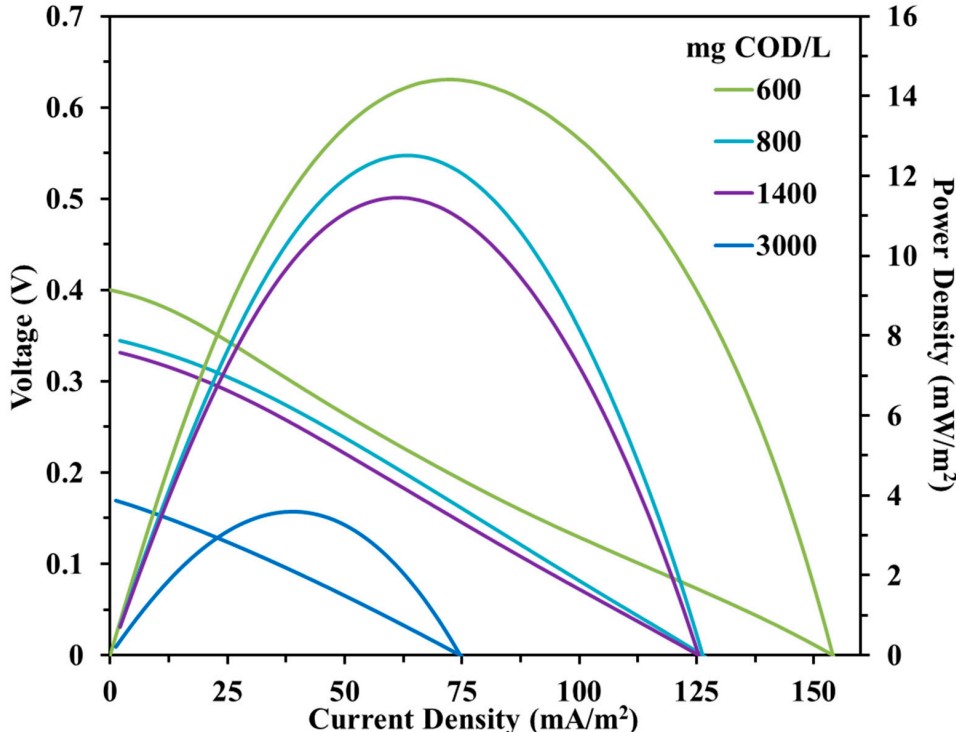

**Figure 6.** Power density versus current density as extracted by LSV experiments on Cell 2 (mullite electrodes) for the following initial condensate concentrations (mg COD/L): 600 ▬, 800 ▬, 1400 ▬ and 3000 ▬.

The highest maximum power output (14.4 mW/m$^2$) for Cell 2 was achieved at 600 mg COD/L. Moreover, the lowest power density (3.6 mW/m$^2$) was obtained for the 3000 mg COD/L initial concentration. For 800 mg COD/L and 1400 mg COD/L the corresponding maximum power densities obtained were 12.5 mW/m$^2$ and 11.5 mW/m$^2$, respectively. Similarly to Cell 1, ohmic losses were exhibited in all cases for Cell 2. Using the power density peak method, the internal resistances were calculated for the corresponding initial condensate concentrations (600 mg COD/L: 91 Ω, 800 mg COD/L: 110 Ω, 1400 mg COD/L: 113 Ω, 3000 mg COD/L: 115 Ω). The internal resistances of Cell 2 presented a similar pattern as for Cell 1. Specifically, by increasing the condensate concentration, the internal resistances increased whereas the maximum power was decreased. Overall, in terms of power output, Cell 2 (mullite assembly) outperformed Cell 1 (GoreTex assembly), indicating the potential of the ceramic material to be effectively used as a separator. In all cases, at low initial condensate concentrations (600 mg COD/L) the maximum power output was high in comparison with the maximum power output values which were obtained at higher initial condensate concentrations (800–3000 mg COD/L). Similar results were previously observed in [35]; by increasing the initial condensate concentration the maximum power output did not present an increase beyond a certain value.

### 3.3. E. Lectrochemical Impedance Spectroscopy Characterization

The results from the LSV experiments indicate that by increasing the initial condensate concentration, the internal resistance of the cells also increased. In order to define the contribution of the different resistances to the total internal resistance, a detailed electrochemical characterization using electrochemical impedance spectroscopy (EIS) measurements was carried out. Figures 7 and 8 present the Nyquist diagrams of Cell 1 and Cell 2, respectively. In both Figures, two distinguishable arcs are presented followed by a Warburg element. The first arc is combined with the biofilm resistance (RBF), while the second arc is attributed to the charge transfer resistance (RCT) as explained by [36], for single-chambered MFCs. RS is calculated by the intersection of the left side of the first arc with the *x*-axis of the Nyquist diagrams.

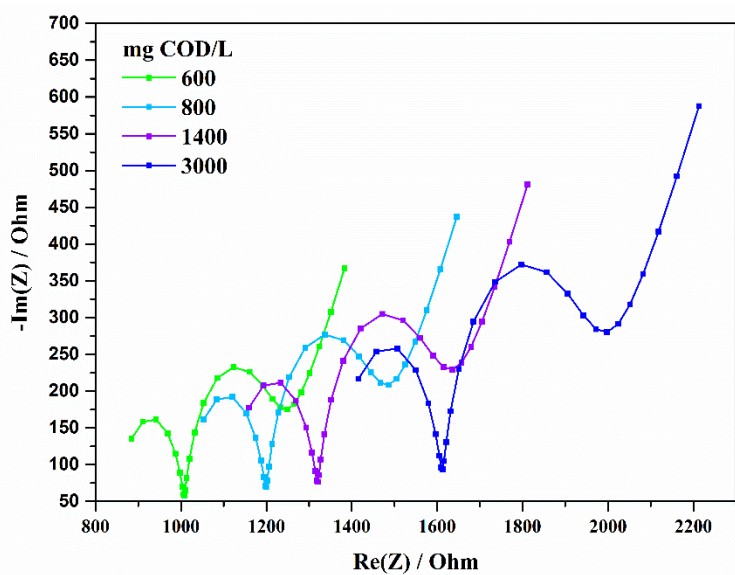

**Figure 7.** Nyquist diagrams of Cell 1 for different initial condensate concentrations (600–3000 mg COD/L).

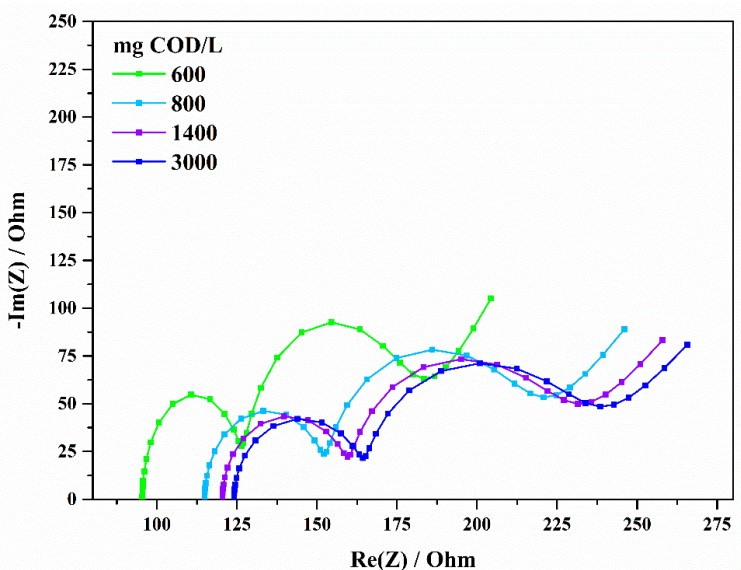

**Figure 8.** Nyquist diagrams of Cell 2 for different initial condensate concentrations (600–3000 mg COD/L).

By applying the fitting model described in earlier publications [4,31], the internal resistances from each EIS experiment were calculated. They are presented in Tables 4 and 5 for Cell 1 and Cell 2, respectively.

**Table 4.** Cell 1 EIS fitted parameters for different initial condensate concentrations.

| Fitted Parameters | mg COD/L | | | |
| --- | --- | --- | --- | --- |
| | 600 | 800 | 1400 | 3000 |
| $R_S$ (Ω) | 330 | 488 | 551 | 741 |
| $R_{BF}$ (Ω) | 349 | 347 | 354 | 349 |
| $R_{CT}$ (Ω) | 119 | 150 | 173 | 293 |
| Rint (Ω) | 798 | 985 | 1078 | 1383 |

**Table 5.** Cell 2 EIS fitted parameters for different initial condensate concentrations.

| Fitted Parameters | mg COD/L | | | |
| --- | --- | --- | --- | --- |
| | 600 | 800 | 1400 | 3000 |
| $R_S$ (Ω) | 19 | 24 | 27 | 30 |
| $R_{BF}$ (Ω) | 31 | 33 | 35 | 34 |
| $R_{CT}$ (Ω) | 45 | 49 | 52 | 56 |
| Rint (Ω) | 95 | 106 | 114 | 120 |

As can be seen from Tables 4 and 5, by increasing the initial condensate concentration the total internal resistance ($R_{in}$) increases for both cells. These results are in accordance with the previous observation that the decrease in the anolyte's pH and conductivity, which occurs with an initial condensate concentration increase, deteriorates the cells' performance. This is reflected in the solution resistance ($R_s$) and the charge transfer resistance ($R_{CT}$) increase for Cells 1 and 2, respectively. On the other hand, the biofilm resistance ($R_{BF}$) is practically constant, which denotes a well-established acclimation procedure on the anodic electrode. In particular, in the case of Cell 1, the $R_s$ and the $R_{CT}$ were 2.5 times higher ($R_s$ increase: 125%, $R_{CT}$ increase: 146%) with the initial condensate increase from 600 to 3000 mg COD/L, whereas $R_{BF}$ was in the range of 347 to 354 Ω. Additionally, in the case of Cell 2, the $R_s$ and the $R_{CT}$ increased by 59% with an initial condensate increase from 600 to 3000 mg COD/L, while $R_{BF}$ was in the range of 31 to 34 Ω.

Furthermore, it is observed that Cell 2 has significantly lower values of $R_S$, $R_{BF}$ and $R_{CT}$ in comparison with Cell 1. This result is correlated with the nature of the cathodic

electrode configuration, since all the other parameters were kept the same. Specifically, in the case of Cell 2 and the mullite assembly, the cathodic electrode exploits the mullite porosity as the exchange medium in which the anolyte infuses and reaches the dense area of the catalytic paste. The whole reaction is enhanced through the significant larger area of the electrode (inner area of the tube) and oxygen reduction is more favored in this electrode in comparison with the GoreTex electrode assembly used in Cell 1. Thus, the lower performance of Cell 1 is attributed to the perforated Plexiglas tubes, which inhibit the full contact of the anodic solution, in contrast to the mullite tubes. Under this consideration, all the values of the resistances are lower in Cell 2.

In particular, $R_{CT}$ is correlated with the overall electron mobility and the enhanced surface area of the cathodic catalyst [36–39], and in the case of GoreTex electrodes a power blockage of the cathodic electrode itself is a restricting factor. In addition, the smaller $R_{BF}$ of Cell 2 is also attributed to the enhanced surface area of the cathodic catalyst, which possibly resulted in the formation of more electrogenic bacteria on the anodic electrode, since microorganisms developed in a better environment. Moreover, the increase in condensate concentration caused a drop in the solution conductivity (Table 2). The measured resistance of the solution, $R_s$, (Tables 4 and 5) was in accordance with that result, since it increased as the condensate concentration increased. As the initial condensate concentration was increased in the feed of the MFC, the pH and the conductivity were reduced. The reduction in the conductivity led to an increase in the resistance of the solution, inhibiting the performance of the MFC. The improvement in the electrochemical performance of the MFC, by increasing the electrolyte conductivity, has been observed before [26]. Moreover, the pH reduction affected the microorganisms which operate better at close-to-neutral pH (~6.5–7) [40]. The LSV and EIS experiments validated the observation regarding the MFC performance. Increasing the initial condensate concentration led to a higher initial COD in the feed, which in the case of Cell 1 resulted in longer operation cycles. In the case of Cell 2, no such increase was observed in the operation cycle duration, this being possibly attributable to the better performance of Cell 2 due to the use of mullite electrodes.

## 4. Conclusions

The operation of two single-chamber MFCs using different cathode assemblies (Cell 1: GoreTex assembly, Cell 2: mullite assembly) was assessed at different food waste condensate concentrations (400–4000 mg COD/L). The work demonstrated that condensate was successfully treated in all cases (COD removal efficiencies >86%). However, the highest values of maximum power output and maximum electricity yield were obtained at lower concentrations, with Cell 2 outperforming Cell 1. In particular, the maximum power density was obtained at 600 mg COD/L initial condensate concentration (Cell 1: 14.2 mW/m$^2$, Cell 2: 14.4 mW/m$^2$) whereas the maximum electricity yield was achieved at 400 mg COD/L for Cell 2 (25.9 mJ/g COD/L) and at 500 mg COD/L (1.51 mJ/g COD/L) for Cell 1. Moreover, the highest current output was 1.6 mA and 0.6 mA for Cell 2 and Cell 1, respectively.

All in all, the results indicate that the mullite cathode assembly exploits the mullite porosity in which the anolyte infuses and reaches the dense area of the catalytic paste, whereas the perforated Plexiglas tube was found to deteriorate this contact. In addition, it is shown that although the decrease in the anolyte's pH and conductivity, which occurs with an initial condensate concentration increase, limits the cells' electrical performance, the condensate is successfully treated in all cases. The RIN values from the EIS experiments corroborate the above results.

These findings indicate that the MFC technology, with further improvement, can be effectively used in the proposed alternative management scenario of HFW. Thus, the MFC can be used after drying and shredding of the HFW for the treatment and exploitation of the condensate by-product originating from this procedure.



**Author Contributions:** Conceptualization, A.T. and G.L.; data curation, A.T., T.K., P.K.P. and G.L.; formal analysis, A.T., T.K. and G.L.; funding acquisition, A.T.; investigation, A.T., T.K. and G.L.; methodology, A.T., T.K., P.K.P. and G.L.; project administration, A.T. and G.L.; resources, A.T. and G.L.; supervision, A.T., C.A. and G.L.; validation, A.T., T.K. and G.L.; visualization, A.T., T.K. and G.L.; writing—original draft, A.T., P.K.P. and T.K.; writing—review and editing, A.T., T.K. and G.L. All authors have read and agreed to the published version of the manuscript.

**Funding:** This research was funded by the Hellenic Foundation for Research and Innovation (HFRI) and the General Secretariat for Research and Technology (GSRT) under grant agreement No. 862.

**Institutional Review Board Statement:** Not applicable.

**Informed Consent Statement:** Not applicable.

**Data Availability Statement:** The datasets generated and analyzed during the current study are available from the corresponding author upon reasonable request.

**Acknowledgments:** This project received funding from the Hellenic Foundation for Research and Innovation (HFRI) and the General Secretariat for Research and Technology (GSRT), under grant agreement No. 862.

**Conflicts of Interest:** The authors declare no conflict of interest.

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
