# Peer review of "Effect of Food Waste Condensate Concentration on the Performance of Microbial Fuel Cells with Different Cathode Assemblies"

_sustainability, doi:10.3390/su14052625_

Round 1
Reviewer 1 Report
This reviewer would like to thank the journal and the authors for the opportunity to review this very interesting manuscript (Manuscript ID sustainability-1555250).
The manuscript addresses a very important topic: the management and chemical treatment of food waste.
This reviewer perceives that the manuscript has scientific value and is well presented.
Next to the value and merits of this manuscript, the reviewer would like to make available the following comment and question to the authors, with the intention that the response would make the value of the manuscript even more explicit.
A comment and two related questions, on this manuscript not having a Discussion section.
Comment:
The Results section (starting in lines 164) includes some discussion of results, however that discussion seems to be scarce. For example, while the manuscript has 47 bibliographic references in the References section, the Results section (that currently includes the discussion of results) seems to refer to references in around only 10 occasions.
Two related questions to the authors
- Could the authors please briefly explain whether the authors would see value in deepening the discussion of results in the manuscript?
- If yes, could the authors please include some more discussion of results in the manuscript?
This reviewer would like to thank the authors in advance for their response.
Reviewer 2 Report
- This paper assessed two types of separators made from two types of materials, GoreTex cloth and mullite. However, the authors did not described clearly why they used these materials and its merits or demerits. Structural characteristics of these two materials have not been provided (Ex. SEM images of the surface and cross section view).
- The process of condensing vapors has not been explained.
- Why at higher COD concentration the operation of the MFCs were not better?
- The surface area of the anode or cathode were not provided. Many references have been cited without discussion about the referred contents.
- Why the authors set the anode as the counter electrode in the LSV measurement?
- Since the results of LSV measurement do not exclude the current generated by the electrode materials, the precision of the obtained power generated by the MFCs is questionable.
Reviewer 3 Report
The manuscript is original, interesting and in my view, it is a great contribution in the field of study. Since it presents A study analyzing the effects of organic load on the performance of an MFC unit. In addition, the authors have worked with different electrodes analysing the differences between both units
In my opinion, the manuscript is highly attractive to other readers and the proposed methodology is clear. The manuscript is well written. The methodology has been well conducted and the conclusions are clearly supported by the results.
I suggest this paper should be accepted but after revisions that can be seen below.
- During the first Bach cycles, the authors used anaerobic sludge, however they do not specify any characteristics. In my opinion, they should include some specification.
Reviewer 4 Report
This paper concerns the performance of microbial fuel cells when food waste condensate is being used as a substrate in different concentrations. The results show high organic removal in all cases, and that the mullite assembly was more efficient. The paper is well written and very precise in terms of materials and methods and the results are well presented.
Author Response
Thank you for reviewing our manuscript.
Round 2
Reviewer 2 Report
The quality of the manuscript has been improved.